# The metazoan landscape of mitochondrial DNA gene order and content is shaped by selection and affects mitochondrial transcription

Noam Shtolz [1] & Dan Mishmar [1✉]

Mitochondrial DNA (mtDNA) harbors essential genes in most metazoans, yet the regulatory impact of the multiple evolutionary mtDNA rearrangements has been overlooked. Here, by analyzing mtDNAs from ~8000 metazoans we found high gene content conservation (especially of protein and rRNA genes), and codon preferences for mtDNA-encoded tRNAs across most metazoans. In contrast, mtDNA gene order (MGO) was selectively constrained within but not between phyla, yet certain gene stretches (ATP8-ATP6, ND4-ND4L) were highly conserved across metazoans. Since certain metazoans with different MGOs diverge in mtDNA transcription, we hypothesized that evolutionary mtDNA rearrangements affected mtDNA transcriptional patterns. As a first step to test this hypothesis, we analyzed available RNA-seq data from 53 metazoans. Since polycistron mtDNA transcripts constitute a small fraction of the steady-state RNA, we enriched for polycistronic boundaries by calculating RNA-seq read densities across junctions between gene couples encoded either by the same strand (SSJ) or by different strands (DSJ). We found that organisms whose mtDNA is organized in alternating reverse-strand/forward-strand gene blocks (mostly arthropods), displayed significantly reduced DSJ read counts, in contrast to organisms whose mtDNA genes are preferentially encoded by one strand (all chordates). Our findings suggest that mtDNA rearrangements are selectively constrained and likely impact mtDNA regulation.

[1] Department of Life Sciences, Ben-Gurion University of the Negev, Beer Sheva, Israel. ✉email: dmishmar@bgu.ac.il

The mitochondrion is central to cellular energy production and metabolism and is present in virtually all eukaryotes. According to the endosymbiotic theory, all mitochondria originate from a single endosymbiosis event that fused a free-living α-proteobacterium and (likely) an Archaean host ~2 billion years ago[1]. Endosymbiosis was followed by a gradual unidirectional transfer of most of the bacterial genes to the host genome (today's nucleus), thus leaving only a few genes in the remaining mitochondrial DNA (mtDNA). For example, whereas human mitochondria require more than ~1000 genes for their activities[2], the human mtDNA contains only 37 genes, encoding for 13 proteins (PCG), all essential oxidative phosphorylation (OXPHOS) subunits, 22 transfer RNA (tRNA) genes, and 2 ribosomal RNA (rRNA) genes[3]. Even though their nucleotide sequences can vary between and even within species[4], previous analyses suggest that mtDNA gene content is conserved across vertebrates[5–7]. Notably, some content differences have been reported among invertebrates, mostly involving tRNA genes[8,9]. Hence, by and large, mtDNA gene content is likely conserved (negatively selected) in most metazoans.

Unlike mtDNA gene content, mtDNA organization and gene order have been given less attention as targets of natural selection. Although most studied organisms harbor a circular mtDNA, some organisms, such as medusozoan, harbor linear mtDNA[10], and in a few metazoans (such as in myxozoa and lice) the mtDNA is fragmented into groups of circular chromosomes that co-segregate across generations[11,12]. Fragmentation of mitochondrial genomes has been previously linked to increased sequence divergence and rearrangement rates[13]. Therefore, one may ask whether and how such dramatic organization changes affected the regulation of mtDNA transcription and replication[14].

Early studies of mtDNA transcriptional regulation in humans[15,16], and mice[17] suggested that the mtDNA is transcribed in strand-specific polycistrones, nearly at the length of the entire linearized mtDNA molecule; such long precursor transcripts are, in turn, cleaved into mature, individual (or di-cistronic) mRNA molecules[18]. Therefore, intuitively, gene order rearrangements within a given polycistron are not expected to affect the local transcriptional patterns or regulation. With this in mind, a previous study of metazoans revealed exceedingly more mtDNA gene order variability in phyla such as Cnidaria, Annelida, and Porifera as compared to other metazoan phyla; furthermore, more than 70% of the studied Mollusca and Annelida possess unique mtDNA gene orders[19]. These findings raise three questions: (A) Is this phenomenon the rule or the exception while considering a broader representation of metazoan phylogeny? (B) Is it possible that the degree of mitochondrial gene order (MGO) conservation, and hence selective constraints over MGO, differs among metazoan phyla? (C) Would changes in MGO affect the regulation of the mitochondrial genome? A recent study of mtDNA-derived nascent RNA transcription in mammals and invertebrates[20] provides first clues to the third question: Precision run-on sequencing (PRO-seq)[21] and global run-on sequencing (GRO-seq)[22] analyses revealed that whereas the studied mammals displayed relatively consistent light strand and heavy strand polycistronic transcripts (as in humans), *Drosophila melanogaster* had multiple initiation and termination sites and *Caenorhabditis elegans* had only a single heavy strand initiation and termination site. Notably, the latter two organisms vary almost exclusively in gene order and location. Therefore, the hypothesis that mtDNA rearrangements affect mtDNA transcriptional patterns is plausible[20].

Previous studies of MGO and mitochondrial gene content were mostly limited to specific taxa such as Aves[7], Arthropoda[23], Nematoda[24], or recently vertebrates[25]. Such analyses focused mainly on using mitochondrial

rearrangements as a tool for phylogenetic comparisons[26,27], assessment of possible recombination rates[25], or reconstructions of ancestral mtDNA gene orders[28,29]. While there are more comprehensive studies performed across Metazoan taxa[19,30], and although they shed new light on the overall diversity and conservation of metazoan mtDNAs, the effect of natural selection on MGO and the functional consequences of mitochondrial rearrangements have not yet been determined. Here, we analyzed the mitochondrial genomes of all available metazoans ($N = 9567$) for mtDNA rearrangements in gene order and assessed the impact of natural selection on such. We then analyzed available RNA-seq across metazoans to assess the impact of mtDNA rearrangements on mtDNA transcriptional patterns.

## Results

### Database construction and validation of metazoan mitochondrial genomes.
To assess the variability of MGO and mtDNA gene contents across metazoan species, there is a need to generate an updated and curated database of all available annotated metazoan mtDNAs. To this end, the annotated mtDNA features of all sequenced metazoan organisms were downloaded, parsed from the NCBI Organelle database[31], and combined with the previously available MitoZoa database[32]. Since we first focused on the analysis of MGO and mtDNA contents, we excluded all features apart from genes from further analyses and unified the gene annotations into a common format across all organisms and databases, while assigning each organism an exact taxonomic lineage. This database contained both the annotation and mtDNA gene order across the mtDNAs of 9657 different metazoans.

Previous work suggested that mitochondrial databases, such as the NCBI Organelle database, suffer from inaccuracies including mistaken topology, incorrect gene denomination, inverse strand specification, erroneous annotations of tRNA, or rRNA genes, non-canonical start codons in protein-coding genes, and partially sequenced genomes[19]. To account for these potential problems, we introduced some quality control (QC) measures to our database construction pipeline, inspired by the MitoZoa database creation process[32]. As a first step, the existence of all listed tRNA genes was validated using a combination of tRNAscan-SE[33] and ARWEN[34] prediction algorithms; organisms with at least one unvalidated tRNA gene corroborated by both algorithms were omitted from the database. In some cases, both algorithms agreed on a tRNA gene that was different from the reported annotation; in such cases, the annotation was amended accordingly (for additional information—see Methods). Second, organisms with fragmented or incompletely sequenced mtDNA or with reported introns (and thus affect the gene order consideration) were also removed from further analyses. These QC filtration steps retained 8053 metazoan organisms, of which most were chordates (64%, $N = 5125$), arthropods (25.2%, $N = 2027$), and mollusks (3.8%, $N = 307$), with the remaining 7% ($N = 594$) spanning 20 different phyla (Fig. 1a). Notably, our analysis nearly tripled any previously analyzed databases[30,32], with ~154% more vertebrates, 426% more arthropods, 204% additional cnidarians and 232% more mollusks (Supplementary Table S1). While most of the new additions expand the numbers of pre-existing taxonomic orders, some add new, previously uncharacterized taxa; there are 17 new taxonomic orders within Chordata, 11 within Mollusca, 8 within Platyhelminthes and 7 within Annelida and Echinodermata (Supplementary Data S1).

### MtDNA gene content analysis across metazoan phylogeny reveals variation in tRNA contents and codon usage bias.
In accordance with previous reports[19], mtDNA gene counts are

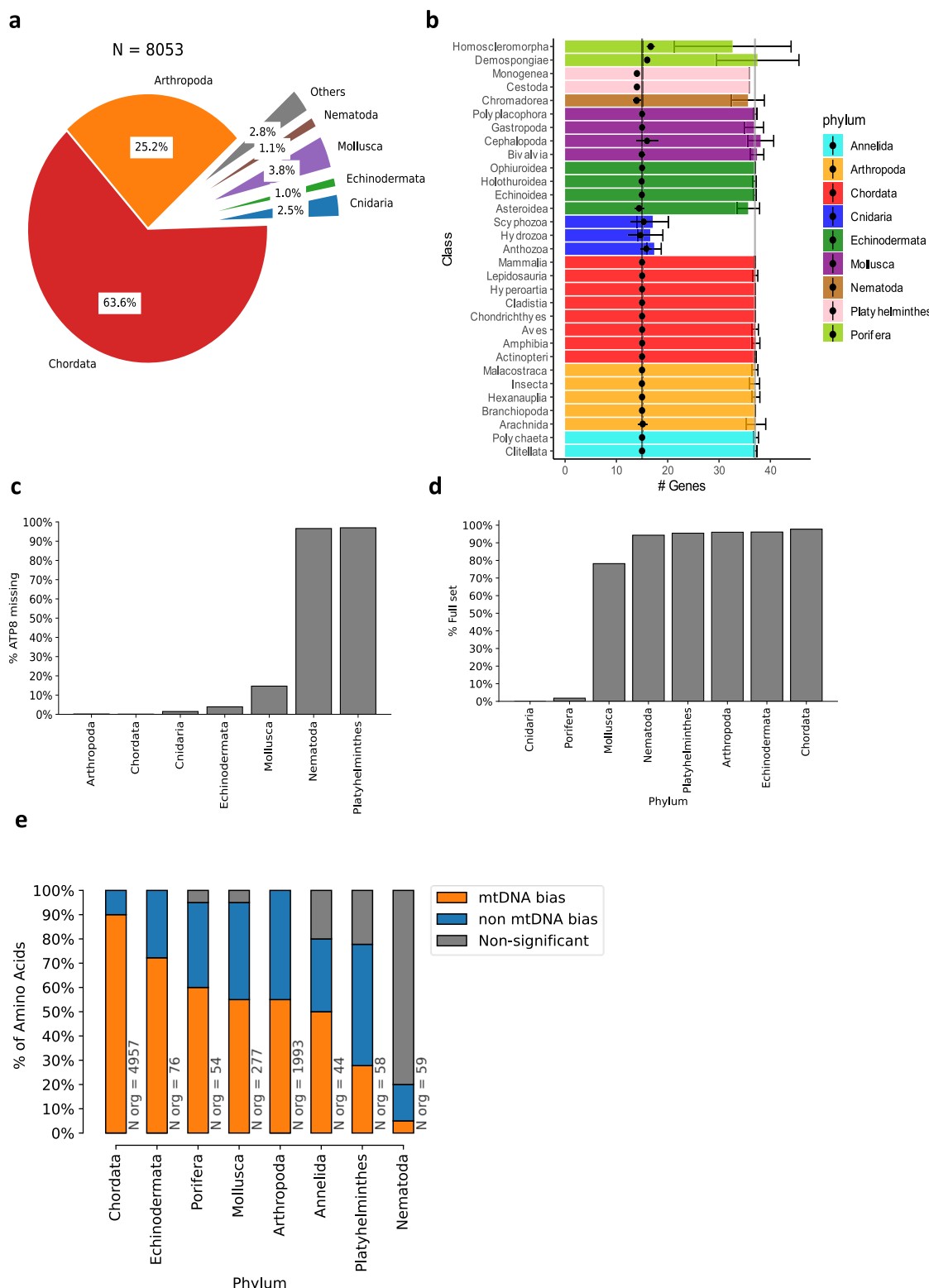

remarkably consistent among metazoan phyla, with an average of $37 \pm 1.4$ (SD) known genes across all metazoans, excluding cnidarians ($N = 7850$). All cnidarians within our database ($N = 204$, 2.5% of all metazoans) have a drastically reduced gene content in comparison to other phyla ($17.2 \pm 1.8$ SD), likely due to an ancestral deletion of most mtDNA tRNA genes except for the mitochondrial *tRNA-Met* and *tRNA-Trp*. The latter tRNA genes are

present in 95% and 61% of the cnidarian species in our database, respectively. mtDNA gene content is highly conserved when one considers only non-tRNA genes: There are typically 15 such genes, including the 12 S and 16 S *rRNA* genes (*rrnS* and *rrnL*), and 13 protein-coding genes (PCGs) that code for subunits of four out of five respiratory chain complexes (*ND1-ND6*, *ND4L*, *CYTB*, *COX1-COX3*, *ATP8*, and *ATP6*) (Fig. 1b). Two noteworthy deviations

**Fig. 1 MtDNA gene content is conserved throughout Metazoa. a** Phylum-level distribution of the 8053 organisms within our analyzed database. **b** Distribution of the number of different mtDNA genes per taxonomic class, colored by phylum. Black dots represent the number of non-tRNA genes; whiskers correspond to the 95% confidence interval range. The grey and black vertical lines highlight the number of mtDNA genes in humans, with or without tRNA genes, respectively. **c** Percentage of organisms within each phylum that lack an mtDNA-encoded *ATP8*. **d** Percentage of organisms within each phylum that contain the human mtDNA-encoded tRNA gene set, termed here 'full set', in the Y axis (i.e., 22 tRNA genes). **e** Summary statistics of codon bias analysis. Each stacked bar shows the percentage of amino acids with two or more possible codons which display either: Significant bias towards mtDNA codons (orange), a significant tendency towards nuclear DNA codons (blue), or nonsignificant (grey). *X* axis—phyla analyzed. See also Supplementary Data S3 and Supplementary Data S4 for raw data (https://figshare.com/projects/Shtolz_2022_mtDNA_evolutionary_rearrangements/156008).

from this common set are *ATP9*, which is present in the mtDNAs of all tested Porifera (sponges), and *ATP8*, which is absent from the mtDNAs of nearly all Nematoda (roundworms), Platyhelminthes (flatworms), and 37% of Mollusca Bivalvia (Fig. 1c). It is important to note that *ATP8* has a highly variable sequence, yet its presence is conserved in its secondary structure. Therefore, we cannot rule out the possibility that the sequence of *ATP8* was heavily mutated and therefore erroneously reported as missing from organisms within the mentioned phyla[35]. Indeed, *ATP8* was putatively identified in the mtDNA of select Nematoda and Platyhelminth species[36,37] and confirmed in several whipworm species[38,39]. Within Bivalvia orders, *ATP8* deletion appears to be limited to a few taxa, namely Mytiloida (61%, $N = 8$), Ostreoida (94%, $N = 17$), and Veneroida (39%, $N = 11$). Notably, the mtDNA of all Unionida samples—the largest sampled Bivalvia order ($N = 40$), harbored *ATP8*, indicating that this gene was lost several times during Bivalvia evolution.

Unlike rRNA genes and PCGs, the repertoire of mtDNA-encoded tRNA genes (termed here mt-tRNA) is more variable both between and within phyla. The most common mt-tRNA set within Chordata (97% of organisms) contains 22 genes, for the translation of 20 amino acids, including two different *tRNA-Leu* genes (of which one recognizes TTR codons, and another recognizes the CTN codons), and two different *tRNA-Ser* genes (of which one recognizes AGY codons and the other recognizes UCN codons). While most organisms across other phyla contain the same set of 22 mt-tRNA genes (Fig. 1d), organisms within taxa such as Cnidaria, Porifera, and Bivalvia consistently deviate from the common set. Organisms that lack the minimum requirement of one mt-tRNA gene per amino acid, will most likely rely on tRNA import from the nucleus to survive. Indeed, evidence for nuclear tRNA import to the mitochondria has been described in Ciliates[40–42], Kinetoplastids[43–45], and even mammals[46], yet this represents the exception rather than the rule, thus suggesting evolutionary preference towards the usage of mt-tRNAs. We, therefore, hypothesized that codon usage in mtDNA-encoded proteins in each organism should be at least partially constrained by the presence of the corresponding mt-tRNAs. To test for this possibility, we measured the relative synonymous codon usage (RSCU, see Methods) for all synonymous codons of each amino acid, while focusing on amino acids with at least two associated codons, of which at least one is recognized by an mt-tRNA (termed here mtDNA codons) and one that is not (Fig. 1e). For most amino acids, Chordata and Echinodermata show a significant bias towards mtDNA codons (90% and 72%, respectively, FDR-corrected *p* value <0.05, Mann–Whitney test, Supplementary Fig. S1a, b). Notably, Mollusca and Arthropoda show only a partial tendency (55% of amino acids, Supplementary Fig. S1c, d), and Nematoda and Platyhelminthes show an opposite bias, towards non-mtDNA codons (Supplementary Fig. S1e, f), i.e., those that are not recognized by mt-tRNAs.

In mammals, protein-coding mtDNA genes are known to display base composition biases as a factor of both strand asymmetry[47] and position within the codon[48]. We thus reasoned

that such biases may confound our results. Prior to correcting for such biases, it is important to note that the heavy and light strand terminology of the mtDNA is a source of confusion and is largely based on the mtDNA nomenclature of mice[49] and human[50]: the heavy strand has a high G + T content as compared to the light strand and is the so-called 'coding' strand[51]. However, that is not the case for other phyla, such as nematodes, in which all genes are located in a single strand, and arthropods which do not have any reported coding strand asymmetry[52]. Therefore, to avoid inaccuracies, we choose to refrain from this terminology throughout the paper and name the strands based only on their directionality (forward and reverse). This nomenclature will be used for our correction of base composition biases (see below).

In order to consider possible base composition biases between the strands, and per codon position, we first assessed such biases across metazoans (Supplementary Fig. S2). Then, we re-performed our analyses while taking these parameters into account, by focusing only on codons that do not contain the most prevalent base within each codon position, per strand, and phylum. Notably, although this stringent filtration unsurprisingly removed most codons, 9 amino acids out of 19 (47%) retain a significant mt-tRNA codon preference in Chordata and 6 out of 10 (60%) in Arthropoda. Additionally, none of the amino acids that have significant biases changed the bias directionality due to our filtration (Supplementary Fig. S3, as compared to S1). Overall, these results suggest a preference for the usage of mt-tRNAs for subsequent translation of mtDNA-encoded PCGs in a phylum-specific manner.

**mtDNA gene organization is under selective constraints across metazoans.** To compare organisms solely based on their mitochondrial gene order and content, we generated a pairwise distance matrix using the common interval rearrangement explorer (CREx) algorithm[53]. Briefly, this algorithm calculates the most parsimonious evolutionary distance based on the minimal number of steps required to radiate the mtDNA gene organization of a given organism from others. In agreement with previous analyses of a smaller metazoan dataset[19], embedding and plotting the distances on a two-dimensional field revealed high conservation within Chordata, with most chordates (69.7%) forming a single, condensed MGO cluster (Fig. 2a). The remaining vertebrates formed another distinct cluster, which is almost entirely composed of birds (i.e., Aves), containing 93.1% of all available birds within our database, which suggests that birds share a unique MGO within Chordata (Fig. 2b). Indeed, 91.4% of the birds within that cluster possess a translocated gene block of *ND6* and *tRNA-Glu* genes (after the *CYTB* gene) as opposed to the main Chordata cluster, in which this block resides between *ND5* and *CYTB* (Supplementary Fig. S4a). Other phyla also display MGO conservation, albeit to a lesser extent than Chordata. For example, arthropods form a single major MGO-based cluster, which can be further divided into several sub-clusters: a large main cluster (50.9% of Arthropoda), and several class-specific sub-clusters (Supplementary Fig. S4b). Overall, these results indicate that

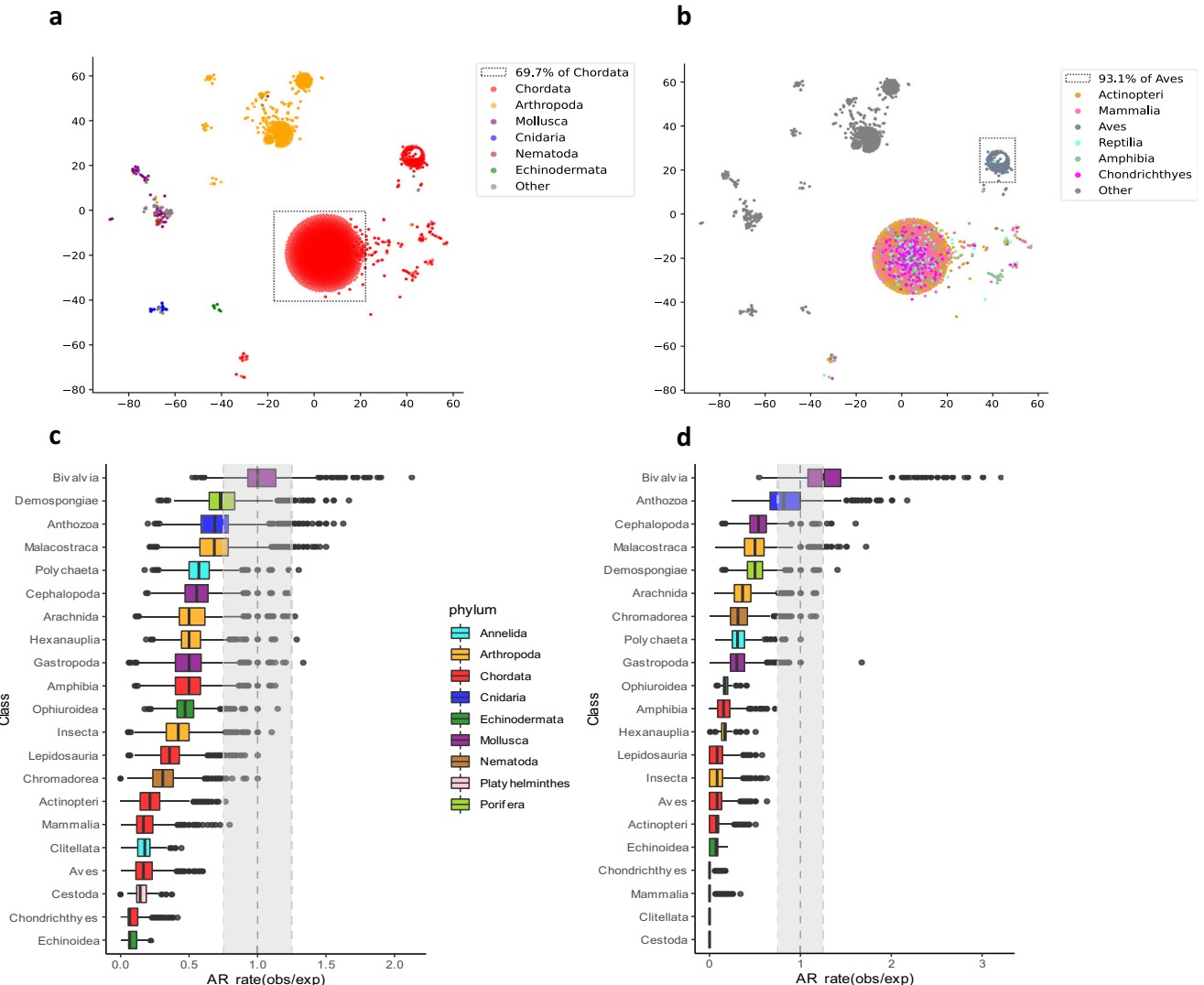

**Fig. 2 MtDNA gene order is more conserved within than between phyla. a**, **b**. tSNE plots of all organisms within the analyzed database, colored either by phylum (**a**) or while highlighting classes within Chordata (**b**). Dotted frame—Chordates in (**a**), birds in (**b**). **c**, **d** Ratio between observed and expected genome architecture (AR rates) within each class, either with (**c**) or without (**d**) mt-tRNA genes. The vertical dashed line and grey rectangle show 1:1 ratio (indicating no difference) and ±0.25 interval around it, respectively. The expected AR-rate distribution was calculated by random sampling of 21 organisms from each class followed by label shuffling 10,000 times. The observed distribution was generated by a similar process but without shuffling. See also Supplementary Data S5 and S6 for raw data (https://figshare.com/projects/Shtolz_2022_mtDNA_evolutionary_rearrangements/156008).

mtDNA gene order and content are under phylum-specific selective constraints, supporting the signature of negative selection.

Next, we inspected MGOs separately for each metazoan class in our database, while using AR rate - a previously defined mitochondrial genome architecture rate metric[19]. In brief, AR rate is defined as the percentage of organisms that share a distinct mtDNA organization. We found that all analyzed Chordata lineages show significantly lower AR rate values than expected by chance (*P* value <0.001, Bonferroni-corrected permutation test, Fig. 2c), supporting the conservation of MGO within chordates, as well as within most other tested classes. Out of all classes, only Bivalvia displayed a significant increase in AR rate compared to the permuted distributions, which is consistent with the reported high mitochondrial genomic variability within bivalves and mollusks in general[19]. The variability is further increased in mollusks due to the unique doubly uniparental inheritance of mitochondria in bivalves which results in further heterogeneity and complexity in their MGO[54]. Notably, the presence/absence

and organization of tRNA genes are the most variable among all mtDNA gene types in metazoans. To estimate the contribution of tRNA genes to the observed AR rate values, we filtered them out and re-performed the AR rate analysis (Fig. 2d). Interestingly, while most classes tended towards conservation (low AR rate values), bivalves retained significantly high AR rate values, which indicates that their mtDNA organizational variability is not exclusively dependent on tRNA genes. Taken together, these results denote that most metazoan taxa tend towards conserved gene order suggesting functional importance.

**The proximity of certain mtDNA gene pairs is highly conserved across metazoans.** Since our analysis suggests that overall MGO is under selective constraints, in a taxa-specific manner, we asked whether shorter gene blocks (i.e., gene pairs, triplets, etc.) are even more conserved, across distant taxonomic groups. To address this question, we generated a prevalence heatmap of all existing non-tRNA gene couples (Fig. 3a). Notably, in contrast to known phylogenetic branching order and proximity, our gene

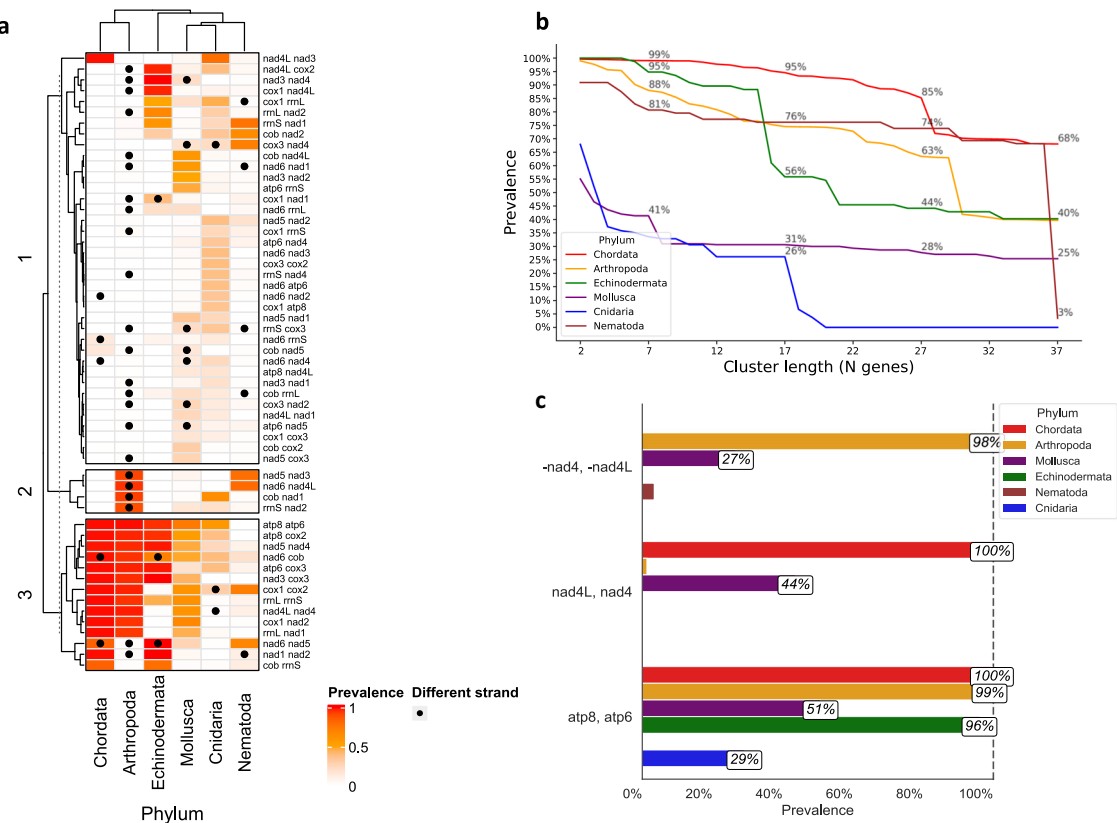

**Fig. 3 Long gene orders are conserved only within taxa, whereas gene couples are highly conserved across metazoans. a** Heatmap showing the pairwise neighboring prevalence (in the percentage of all observed gene couples by phylum). Black dots indicate that the indicated two genes are in opposing strands. Rows and columns are hierarchically clustered based on Euclidean distance. **b** The prevalence of the most common gene cluster within each phylum for every cluster length (2–37 genes). **c** The per-phylum prevalence (in percentage) of the gene couples *ATP8 ATP6*, *ND4L ND4* (forward strand), and *-ND4 -ND4L* (reverse strand). See also Supplementary Data S7 and Supplementary Data S8 for raw data (https://figshare.com/projects/Shtolz_2022_mtDNA_evolutionary_rearrangements/156008).

pair similarity analysis revealed that Chordata clustered closer to Arthropoda than to the evolutionarily closer Echinodermata phylum[55]. While investigating the conservation of gene proximity for larger gene groups (2–37), we found that Chordata contains the most conserved gene block for all gene block sizes (Fig. 3b). Additionally, we found certain gene couples whose proximity was highly conserved across phyla, especially *ATP8-ATP6* and *ND4L-ND4*. The first gene couple, *ATP8-ATP6*, remained adjacent in the forward mtDNA strand of nearly all Chordata, Arthropoda, and Echinodermata (99%, 98%, and 96%, respectively). While the second gene pair, *ND4L-ND4* mapped to the mtDNA forward strand across all Chordata, this gene couple inverted into the mitochondrial reverse strand in nearly all Arthropoda. This phenomenon most probably reflects an evolutionarily ancient inversion that likely occurred in the last common ancestor of all arthropods (Fig. 3c). This indicates that, whereas conservation of relatively long mtDNA gene order occurs within phyla, conservation of gene couple proximity can be much deeper across metazoan evolution, again reflecting the signature of negative selection.

**Evolutionary changes in mtDNA order affect mtDNA gene expression: the case of alternating gene blocks**. Since the mtDNA of all tested metazoans is transcribed in polycistrones, we hypothesized that the conservation of long gene blocks reflects negative selection acting on the regulation of the entire mtDNA, whereas the conservation of gene couples, especially that of *ATP6-ATP8* and *ND4-ND4L*, whose reading frames uniquely

overlap[56], may reflect negative selection acting on post-transcriptional regulation. Nonetheless, the conservation of gene blocks with intermediate length is not as easily interpreted (Fig. 3b). Firstly, it is logical that the inversion of a gene block from one mtDNA strand to the other will naturally lead to the inclusion of such gene block in the opposite strand's polycistron. In consistence with this thought, we have previously shown, using global precision run on transcription (PRO-seq), that strand shifts between gene blocks not only changed their coding strand but also mark the end of polycistronic transcriptional units in *D. melanogaster*[20]. Therefore, we asked whether this finding is limited to the tested organisms, or whether it marks an alteration in mtDNA transcriptional regulation in organisms sharing the same rearrangements.

A direct way to address this question would be via analysis of either PRO-seq[21] or GRO-seq[22] data, which are both designed to selectively isolate nascent RNA, and enable the identification of transcription start and termination sites in both the nuclear and mitochondrial genomes[20]. In contrast to PRO-seq, RNA-seq measures the steady-state levels of transcripts and hence harbors both nascent and mature RNA in the tested sample. As RNA-seq data is publicly available from a variety of metazoans, we asked whether it could be used to calculate the levels of strand-specific polycistrones. To this end, we calculated read density in mtDNA regions encompassing gene-gene junctions within the same strand (SSJ) as compared to the read density of junctions between genes encoded by two mtDNA strands (DSJ) (for details—see Methods) (Fig. 4a). Specifically, we counted reads within approximately two-reads-long windows around each junction

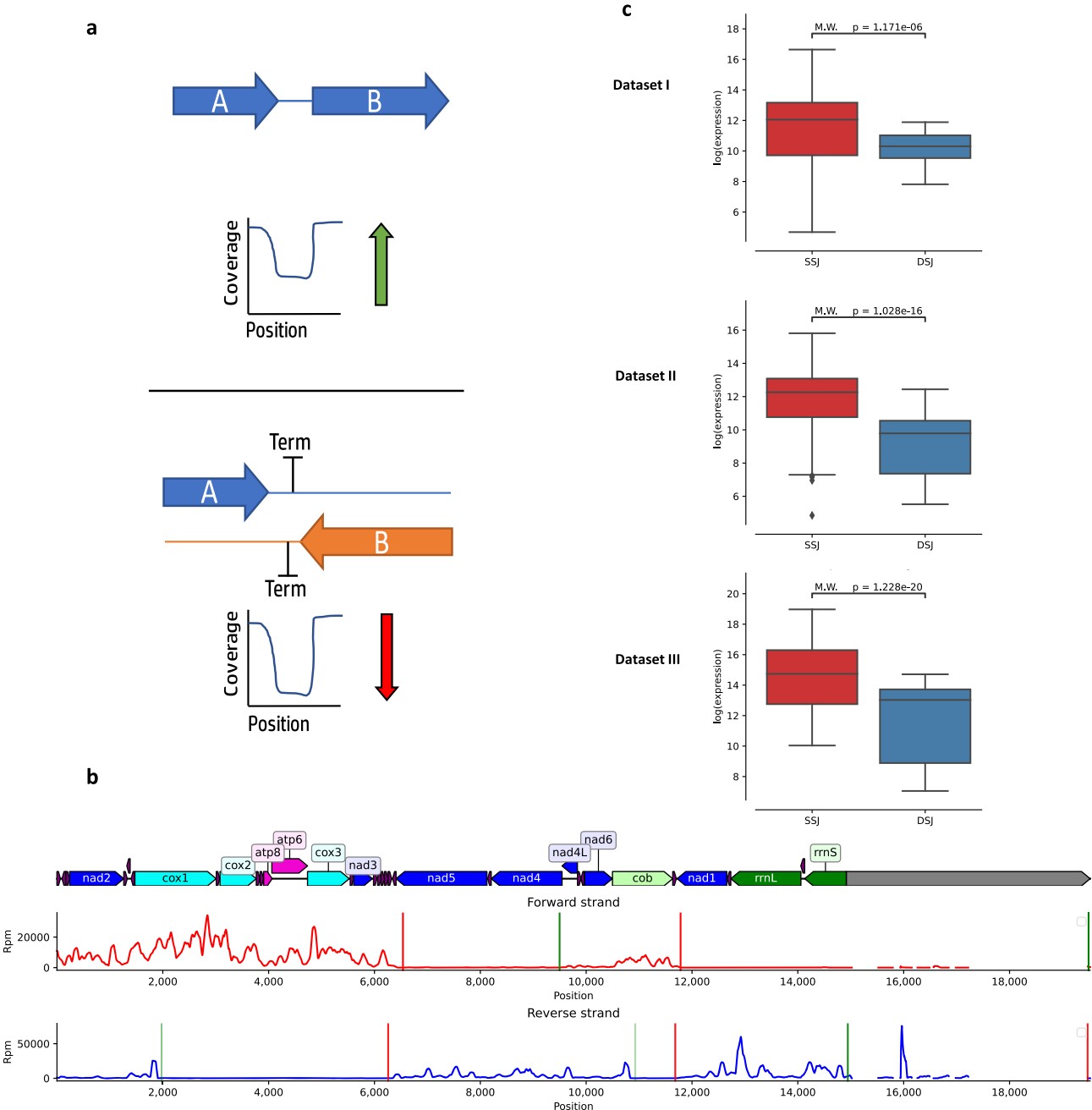

**Fig. 4 Gene-gene junctions represent borders between polycistronic units in Drosophila melanogaster. a** An illustration of the expected expression level in junctions between genes from the same strand (SSJ) as compared to junctions between genes encoded by different strands (DSJ). **b** Analysis of mtDNA sequence read coverage from precision run-on sequencing (PRO-seq) experiments in Drosophila. Forward strand read coverage is colored in red and the reverse strand in blue. Vertical green and red lines mark the predicted transcriptional initiation and termination sites, respectively. The level of vertical line transparency depends on the prediction's confidence score (see Methods). The numbers below each line correspond to mtDNA positions. **c** Boxplot comparisons of log normalized expression between DSJ and SSJ in three different Drosophila RNA-seq datasets (See Supplementary Data S2). Each box extends from the 25th to the 75th percentiles, whiskers denote values within 1.5 interquartile range of the percentiles. Dots denote outliers outside of the mentioned ranges. M.W. Mann–Whitney test values. SSJ vs. DSJ effect sizes (Cohen's *d*) per-dataset: 0.67, 0.95, and 0.339. See also Supplementary Data S9 and Supplementary Data S10 for raw data (https://figshare.com/projects/Shtolz_2022_mtDNA_evolutionary_rearrangements/156008).

(between 50–150 bases, depending on the RNA-seq library construction protocols); the windows were designed to overlap both the 3' or 5' end of a given gene and the non-coding junction. As most available RNA-seq data did not discriminate the transcripts according to their strand of origin, reads were counted irrespective of their strand, normalized using TPM, and compared across samples and organisms. We generated a pool of 1037 samples from 77 different metazoan organisms. To minimize sample size effects, organisms with less than five available samples were excluded from further analyses. Secondly, we filtered out junctions between neighboring genes with low expression levels, i.e., their RNA-seq read coverage was lower than two standard deviations from the sample's mean mtDNA gene expression levels. Third, some library selection methods

(such as poly-A selection), likely enrich for reads originating from mature mRNA and filter out the pre-mRNA transcripts we are interested in, resulting in low to non-existent intergenic expression regardless of polycistronic transcription. To avoid artifacts, we chose to avoid poly-A and expression sequence tag type enrichments as much as possible and excluded datasets where over 50% of the identified junctions displayed zero read coverage. After applying these quality control measures, we were left with 950 samples from 53 different metazoan species (Supplementary Fig. S5).

To determine whether the analysis of RNA-seq junction expression data can provide valid information on the transcriptional map, we chose to start with the well-studied model organism with a previously analyzed mtDNA transcription pattern—*D. melanogaster*[20]. mtDNA transcription in Drosophila is initiated at five confirmed initiation sites, of which two are within the reverse strand, and three within the forward strand. Using a modified version of a previously established calculation approach[20], we analyzed the transcription start sites (TSS) and transcription termination sites (TTS) in the Drosophila mtDNA using new publicly available PRO-seq data[57] and confirmed the known transcriptional pattern of Drosophila mtDNA (Fig. 4b). While analyzing RNA-seq data from three unrelated Drosophila samples, different-strand junctions (DSJ) were found to be significantly less expressed than same-strand junctions (SSJ) across all tested datasets (*P* value <0.001, Mann–Whitney test, Fig. 4c), thus supporting our hypothesis. Hence, and as evident by the PRO-seq results, and previous work[20], junctions surrounded by two genes on different strands harbor a TTS and represent borders between two different strand-specific polycistronic transcripts in Drosophila mtDNA. However, while analyzing human mitochondrial transcription, which is composed of two, well-established strand-specific polycistronic transcripts, that encompass all mtDNA genes, we did not find any significant trend toward lower coverage in DSJ compared to SSJ (Supplementary Fig. S6a). This most likely stems from the fact that, unlike Drosophila, in human mtDNA, each strand is transcribed into a single transcriptional unit that encompasses all the genes in that strand, thus transcribing through all gene-gene junctions (Supplementary Fig. S6b).

While inspecting mtDNA gene orders, we noticed that ~90% of the arthropod species display an mtDNA organization that is like in Drosophila: i.e., co-oriented groups of two or more non-tRNA genes that continuously alternate between the two mtDNA strands. We henceforth term this organization alternating gene block organization. Interestingly, alternating gene blocks organization is also found in the mtDNAs of 28.6% and 26% of Cnidarian and Mollusca, respectively (Fig. 5a). We thus asked whether such organisms display similar DSJ-SSJ differences in read coverage as in Drosophila. Out of 31 organisms with such organization and available RNA-seq data (Fig. 5b), 55% (*N* = 17) displayed significantly reduced read coverage around DSJ (Fig. 5c, Bonferroni-corrected *p* value <0.001). In contrast, out of 24 organisms that did not display alternate gene blocks organization, only 16.6% (*N* = 4) showed significantly reduced read coverage around the DSJ (Fig. 5d, Bonferroni-corrected *p* value <0.002), suggesting that species with alternating gene blocks organization had a higher tendency towards a lower expression of DSJ regions as compared to organisms with no alternating gene blocks organization (*p* value = 0.001, $\chi^2$ test of independence). Furthermore, nearly all organisms with alternating gene blocks organization displayed a trend towards reduced expression around DSJ (albeit not all with significant values), a trend that was not observed in species lacking alternating gene blocks organization. Overall, these results provide the first evidence for a functional connection between evolutionary rearrangements in mtDNA organization and changes in the mtDNA transcriptional scheme.

**Different strand junction loci are enriched for motifs in species with alternating gene block organization**. If gene blocks mark the boundaries of mtDNA transcriptional units in arthropods, it is expected that such boundaries will be bound by transcription factors that will mediate transcriptional initiation from one end, and termination from the other end. In humans, it has been discovered that the transcription factors *TFB2M*, *POLRMT*, and *TFAM* form the core of the mitochondrial transcription initiation complex, while *mTERF1* (and possibly additional members of the *mTERF* family) mediate transcriptional termination[58]. Although protein orthologues for these factors have been identified in multiple metazoans[59–61], little is known about the function of the mitochondrial transcriptional machinery in non-human species, especially invertebrates, including arthropods. As a first step towards a mechanistic analysis of the borders of arthropod transcriptional units, we hypothesized that species with alternating mtDNA gene block organization provide an excellent opportunity to identify sequence motifs within the block-block boundaries. Therefore, we searched our metazoan database for sequence motifs enriched in the regions of block-block boundaries in arthropods with alternating gene block organization. Specifically, we screened for sequence motifs located not >500 bases away from either side of the DSJ. We found a total of 39 such motifs that were significantly enriched in species with alternating gene block organization (FDR-corrected *E* value <0.05). Five of these motifs were near DSJs (Fig. 6a), three within tRNA genes between *ND3* and *ND5*, one between *trnP* and *trnT*, and another motif within *ND1*, yet 378 bases away from the DSJ between *ND1* and *CYTB* (Fig. 6b). Importantly, the *ND1* motif was identified in a relatively non-conserved region at the protein level, yet conserved at the DNA level, which supports its possible regulatory function (Fig. 6c). Notably, *dmTTF*, an mTERF-like termination factor has been shown to bind intergenic junctions in Drosophila mtDNA in-vitro and to recognize AT-rich motifs[61]. Most interestingly, we found three sequence motifs (indexed as motifs 2, 4, and 5) that are all within 50 bases of the putative binding region and 300 bases of the predicted transcription termination position. Additionally, the identified five motifs form an AT-rich consensus sequence which is 54.5% similar to the reported *dmTTF* binding sequence in Drosophila. Taken together, these results provide an initial basis for the functional mechanism underlying the alternating gene block organization.

## Discussion

In this study, we analyzed the dynamics of both mitochondrial gene content and gene order across the evolution of over 8000 metazoans. Firstly, our analysis of mtDNA sequences across all available metazoan species indicates that mtDNA gene content is largely conserved while considering the PCGs and rRNA genes across metazoans. Nevertheless, while considering tRNA genes, variation does occur in cnidarians which lost nearly all their mtDNA-encoded tRNA genes, in addition to high tRNA content variability in sponges and mollusks. Despite the variability in these three phyla, the general tendency is to retain tRNA gene content across metazoan mtDNAs, thus suggesting selective preference towards their usage during the translation of mtDNA-encoded proteins. Indeed, we found preferential usage of codons recognized by mtDNA-encoded tRNAs in Chordata, Echinodermata, and Porifera, albeit to a lesser extent in Mollusca, Annelida, and Arthropoda. Interestingly, this preference is virtually lost in Platyhelminthes and Nematoda. Although such preference for the usage of mtDNA codons was previously shown

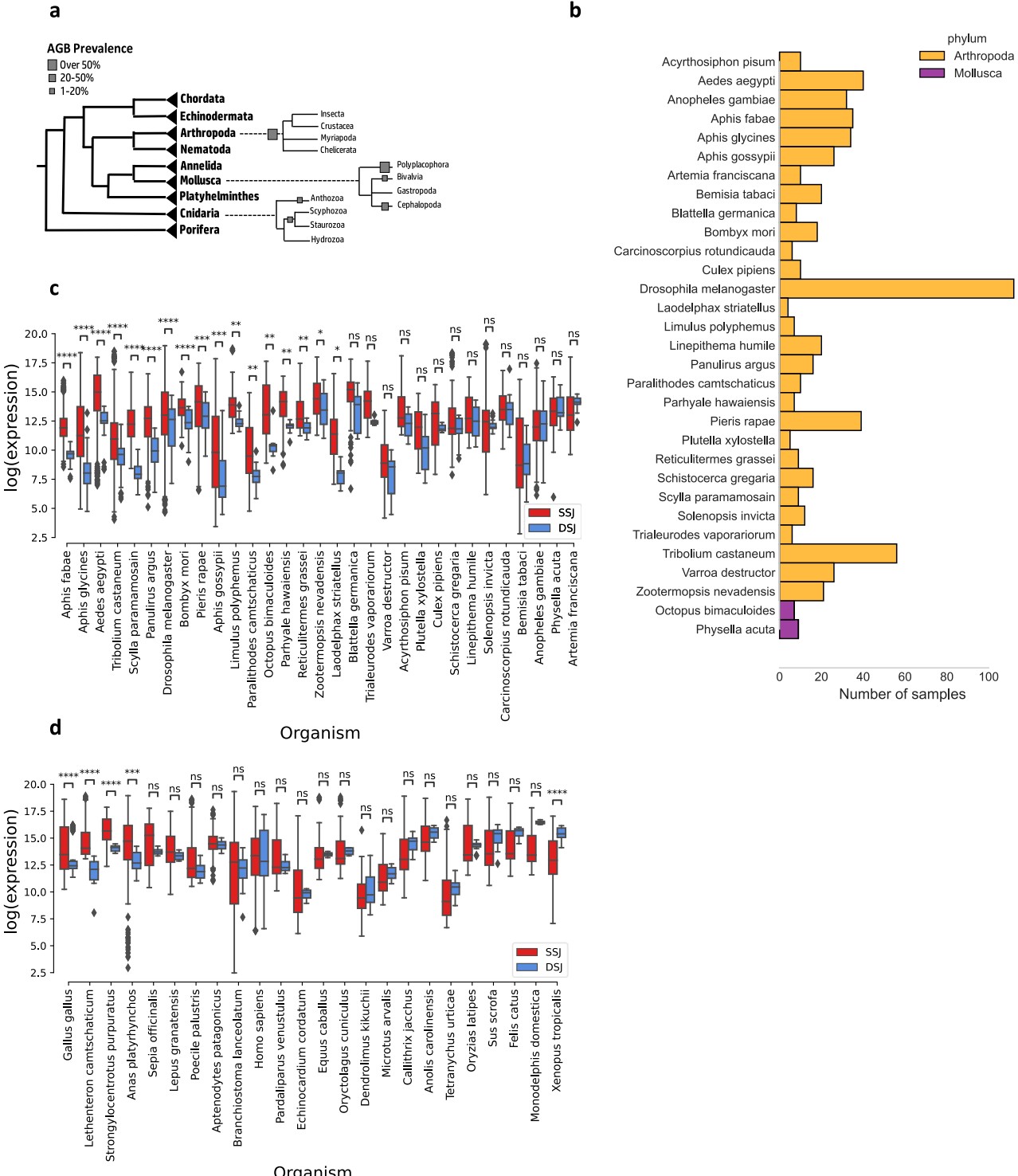

**Fig. 5 Significantly reduced DSJ expression in species with alternating mtDNA gene blocks organization. a** Percentage of different species with alternating gene blocks organization within each phylum. **b** Number of analyzed RNA-seq samples per species for arthropods with alternating gene block organization. **c**, **d** Boxplot comparisons of log normalized expression between different strand junctions (DSJ) and same strand junctions (SSJ) in species with (**c**) or without (**d**) alternating gene block mtDNA organization. *P* value ranges are marked by asterisks. The *p* values were calculated using a two-sided Mann–Whitney test, corrected for multiple testing using the Bonferroni correction. * $0.01 < p \leq 0.05$, ** $0.001 \leq p < 0.01$, *** $0.0001 < p \leq 0.001$, **** $p \leq 0.0001$. Each box extends from the 25th to the 75th percentiles, whiskers denote values within 1.5 interquartile range of the percentiles. Dots denote outliers outside of the mentioned ranges. See also Supplementary Data S10 for raw data (https://figshare.com/projects/Shtolz_2022_mtDNA_evolutionary_rearrangements/156008).

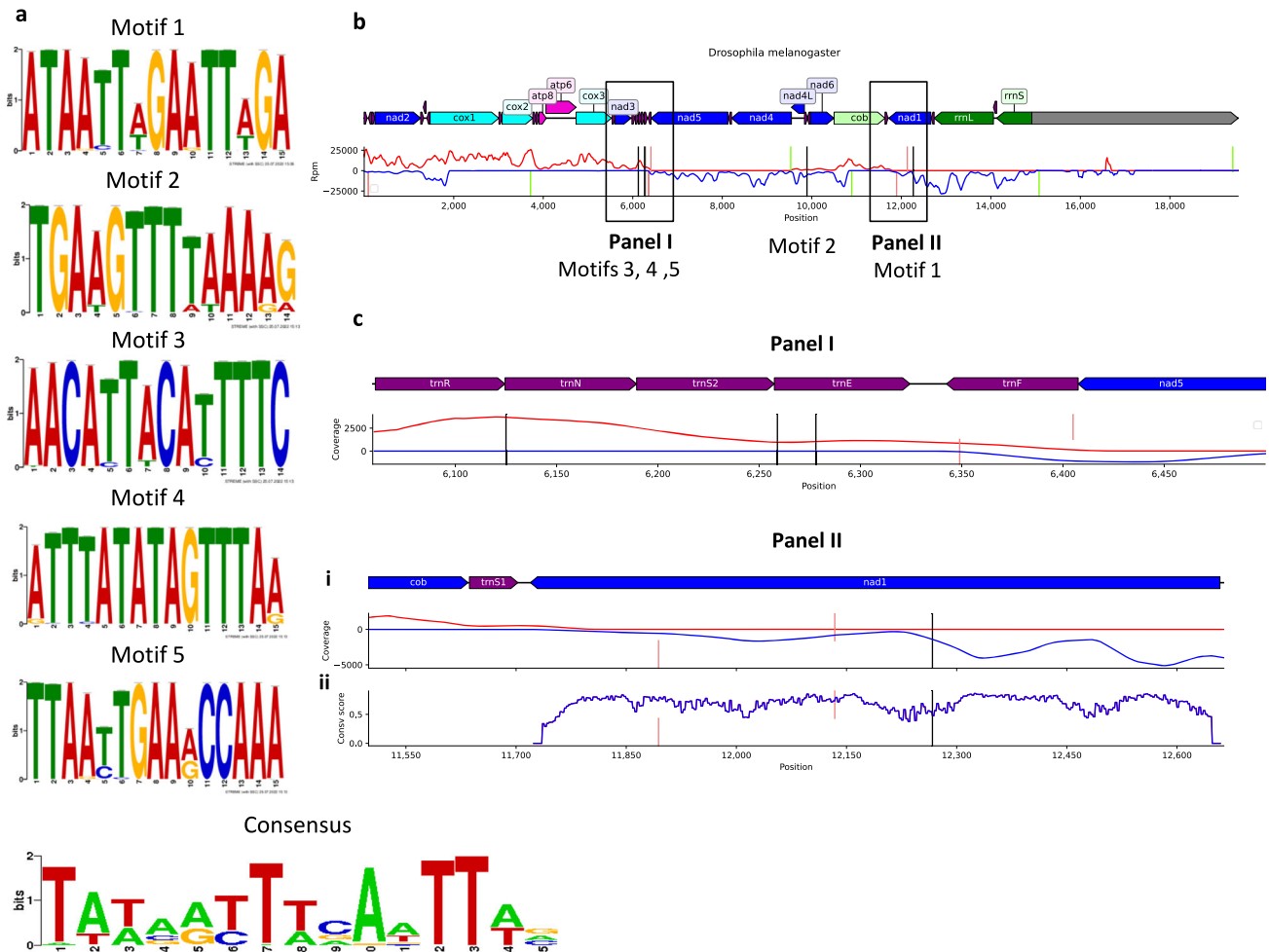

**Fig. 6 Different strand junction loci contain conserved sequence motifs that are enriched in species with alternating gene block organization.** **a** Sequence logo plots of each of the five motifs that are significantly enriched in alternating gene block organization species, and of their consensus sequence. **b** Coverage plot of an example *D. melanogaster* PRO-seq sample. The forward strand coverage is shown in red and the reverse strand coverage in blue. Motifs are marked as vertical black lines, and transcription initiations and terminations are marked as vertical green and red lines, respectively. The motifs' numbering appears below the vertical black lines. **c** Zoomed-in coverage plots showing the positions of motifs 3, 4, and 5 (Panel I) and motif 3 (Panel II). The bottom plot within Panel II (plot ii) shows the protein conservation score per codon of *ND1*. See also Supplementary Data S11 for raw data (https://figshare.com/projects/Shtolz_2022_mtDNA_evolutionary_rearrangements/156008).

for humans[62] and several other animals[63,64], in this study we extended this observation to the entire metazoan tree, and argue that such preference is phylum specific, yet ancient. Additionally, the bias holds, for most amino acids, even after limiting the analysis to codons with less-prevalent bases in each codon position. This suggests that our results may be independent of other confounding factors that are known to affect nucleotide frequencies, such as strand-specific composition differences[47], context-dependent mutations[65], and tRNA and PCG gene expression[62]. Admittedly, our imposed limitation, while stringent, may not eliminate all factors that affect mtDNA base composition. A future comprehensive study of the evolutionary forces that shape mtDNA base composition across all available metazoans may allow for more precise mt-tRNA bias analyses[48]. With that said, how could one explain the reduction, and even lack preference towards usage of mtDNA codon in some taxa? Firstly, the tRNA repertoire variation in cnidarians, mollusks, sponges, and roundworms suggests that these phyla evolved to allow tRNA import from the cytosol into the mitochondria to enable the translation of mtDNA encoded proteins within the mitochondria using the highly conserved mitochondrial ribosome. As tRNA import has been previously identified across the tree of life[40–46],

these phyla may serve as future models to study the phenomenon. Other non-mutually exclusive possibilities are that the reduction in mtDNA codon bias in Mollusca and Arthropods, and the absence of such bias in other taxa, is either due to relaxation of constraints or due to the possibility that mitochondrial tRNAs may recognize more than a single codon thanks to post-transcriptional modifications[66–69]. If the latter constitutes a conserved mechanism across metazoans the preference towards the usage of mt-tRNAs might be even greater than we identified in the current study. This requires further analysis in the future.

Unlike mitochondrial gene content, mtDNA gene order has been previously argued to be less conserved in certain phyla, and hence under lesser selective constraint[19]. Nevertheless, while assessing the dynamics of MGO, we found MGO conservation within taxa, consistently with the phylum-specific signature of negative selection, thus reflecting possible functional implications to mtDNA regulation. This finding is not intuitive, especially for the study of mtDNA transcriptional regulation: the mammalian mtDNA is transcribed in its entirety as a single RNA precursor transcript, per mtDNA strand[70,71]. In the current study, we showed by RNA-seq analysis that the mtDNA of most arthropods, which share the human mtDNA gene content, but profoundly differ from

mammals in possessing alternating mtDNA gene blocks, tend towards reduced RNA-seq read density in gene-gene junctions from different strands (DSJ), as compared to junctions between genes from the same strand (SSJ). This finding suggests that, unlike chordates, DSJs in organisms with alternating mtDNA gene blocks most probably mark the end of polycistrones. This interpretation is supported by previous analysis of nascent mitochondrial RNA transcripts from *D. melanogaster*, explicitly showing that the beginning (5') and end (3') of mtDNA gene blocks also also mark the transcription start and termination sites, respectively[20]. This suggests that mtDNA rearrangements that occurred during evolution, affect mtDNA transcriptional patterns, which tend to be shared by organisms with the same mtDNA organization. Although mtDNA organization in alternating gene blocks is prevalent, especially in arthropods, it can also be found in other groups (certain cnidarians, for instance), thus raising interest in investigating patterns of mtDNA transcription in such organisms. Notably, our analysis indicated that metazoans whose mtDNA is not arranged in alternating gene blocks, i.e., most genes are encoded by one of the strands, did not display significant sequence coverage differences between gene-gene junctions in different strands versus same-strand gene-gene junctions. These findings again support the thought that those differences in the organization of the mito-chondrial genome during evolution likely affect the mtDNA transcriptional pattern (Fig. 7). Nevertheless, one cannot rule out the functional involvement of factors other than transcription. Indeed, mtDNA transcription is known to be heavily coupled to replication in mammals, with both processes relying on the same RNA polymerase, used for both primer synthesis and transcription[58,72]. However, organisms with alternating gene block organization like Drosophila contain additional initiation points, outside of the mitochondrial non-coding region and the origin of replication, and thus contain transcription initiation points that are most likely decoupled from replication. Taken together, one cannot easily deduce from the observed mtDNA transcriptional pattern in a certain organism the entire metazoan phylogeny. Thus, future analyses of mtDNA transcription across the metazoan phylogeny are imperative.

What might be the mechanism which underlies the impact of altered MGOs on the mtDNA transcriptional pattern? One may consider the existence of mtDNA transcriptional regulatory elements throughout the mtDNA. Indeed, in addition to the con-sensus transcriptional regulatory elements, which are mainly located in the non-coding mtDNA region—the D-loop, accumu-lating pieces of evidence point towards transcription factor binding sites throughout the mtDNA[14,73] of which some were shown to participate in the regulation of mtDNA transcription[14]. It is thus possible that like chromosomal aberrations in the nucleus, a change in the location of the binding sites of transcription factors due to mtDNA rearrangement will affect the scheme of interactions between the transcription factors and the mtDNA transcriptional machinery. If this is the case, then the map of mtDNA transcription factor binding sites should differ among metazoans. As an initial step toward testing this thought, we discovered sequence motifs that are significantly enriched in species with alternating gene block organization, some of which were proximal to the in-vitro binding location of a known termination factor in *D. melanogaster*. The availability of genomics techniques that identify protein-DNA interaction sites such as chromatin immune precipitation-sequencing (ChIP-seq) and methods that detect the landscape of chromatin accessibility (ATAC-seq/DNase-seq) may enable the identification of such mtDNA binding sites during evolution. The first glance into such was previously provided by analysis of ATAC-seq and DNase-seq data in humans and mice showing the gradual formation of higher-order mtDNA organization in both mammals[74], involving conserved occupancy patterns[75]. We spec-ulate that the implications of such techniques to a variety of organisms representing metazoan phylogeny may shed new light on the functional impact of mtDNA evolutionary rearrangements on mtDNA organization.

Finally, we discovered that certain gene couples, especially *ATP6-ATP8* and *ND4L-ND4*, retained their proximity across most metazoans. Since there is an open reading frame overlap within each of these gene couples, it is logical to hypothesize that patterns of mtDNA translation of these gene couples are expected to be extremely conserved in metazoans. Such a prediction could be tested experimentally if methods such as ribosome profiling

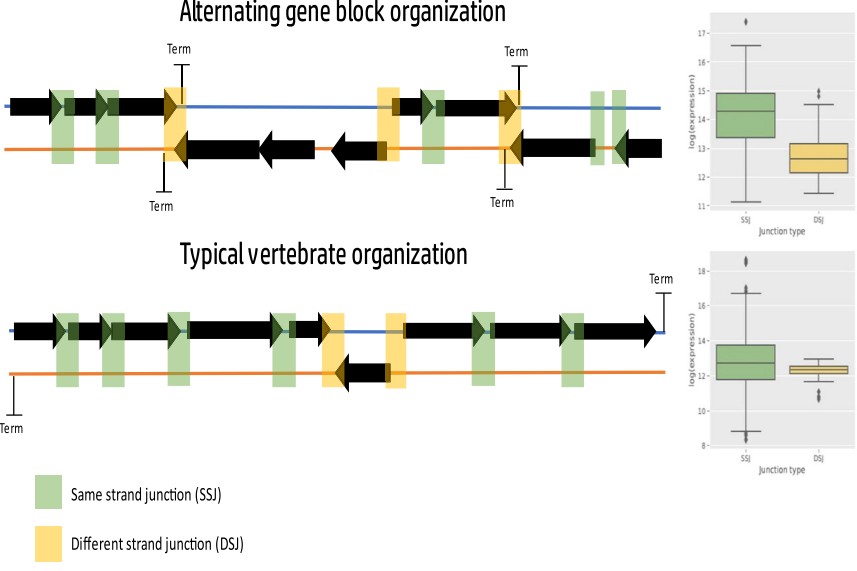

**Fig. 7 Species with alternating gene block organization show a pattern of lower expression around DSJ.** Illustration showing that transcription termination sites coincide with different strand junctions (DSJ) in organisms with alternating gene block organization. DSJ is in yellow, same strand junctions (SSJ) are in green. The semi-transparent rectangles mark the junction location, colored according to type. The boxplots on the right side present an example of the typical expression difference found in organisms with (upper panel) or without (lower panel) alternating gene block mtDNA organization.

will be employed to study mitochondrial translational patterns in representative organisms across metazoan phylogeny.

In summary, our analysis of the largest compendium of metazoan mtDNA sequences assembled to date revealed gene content conservation across most metazoan species, yet variability in the number and nature of tRNA genes. This analysis also showed a preference for mtDNA-encoded protein genes towards usage of codons recognized by mitochondrial tRNAs, suggesting co-evolution between such preference to tRNA gene presence/loss dynamics. Secondly, and most strikingly, our analysis of mitochondrial gene order revealed phylum-specific conservation of long gene blocks, implying the signature of negative selection due to functional importance. Our analysis of available RNA-seq data suggested that in organisms with alternate mtDNA gene block organization (mostly arthropods), the gene block boundaries likely mark the boundaries of polycistrones, in contrast to organisms with other mtDNA organizations. These results, along with previous analyses of nascent mitochondrial RNA transcripts, provide the first clues suggesting that altered mtDNA gene organization during evolution corresponds to changes in transcriptional patterns. This paves the path toward future studies of nascent mitochondrial RNA transcription across the entire metazoan phylogeny. Such a future study will potentially identify transcription initiation, pausing, and termination sites in a variety of organisms, not only in model organisms, and may enable the identification of novel mitochondrial regulatory elements. Additionally, a better understanding of mtDNA regulation will also be important for future studies of coordinated mtDNA-nuclear DNA gene expression in health, disease, and evolution[76,77].

## Methods

**Creation of a database of mitochondrial DNA features in metazoans.** To construct a comprehensive database of metazoan mitochondrial gene features (i.e., annotated genes, and their described mtDNA genes order) of as many organisms as possible, all available mtDNA features were extracted and combined from both the NCBI Organelle database[31] and the MitoZoa database[32] (v2.0.0), yielding the complete mtDNA features of 9657 different metazoan organisms. Python package Biopython[78] (v0.5.0) was used to extract additional information from the national center for bioinformatics (NCBI), and pandas package[79] (v1.3.1) was used to manipulate and combine the two datasets. In brief, the scripts were written to retrieve each organism's gene information and taxonomic lineage based on their unique NCBI RefSeq ID, assembled all gene symbols into a single format, and mark organisms with either incomplete or fragmented mtDNA, with coding sequence (CDS)-containing introns or with unvalidated tRNA genes. Notably, such incomplete/fragmented mtDNAs were excluded from further analysis; for the sake of simplicity, mtDNAs with introns were also excluded. tRNA validation was applied to all identified tRNA genes using two different algorithms, tRNAscan-SE[33] (v2.0.5) and ARWEN[34] (v1.2.3); both algorithms locate and denominate tRNA genes based on their respective DNA sequence. The tRNA validation script can mark a given tRNA as validated if either one of the algorithms confirms the presumed annotation; a tRNA was marked 'unvalidated' if either (A) both algorithms did not locate a tRNA, (B) the two algorithms did not agree on the denomination, or if (C) both algorithms agreed on tRNA annotation which was different from the reported one. Since organisms with at least a single unvalidated tRNA, incomplete or fragmented mtDNA, or CDS-containing introns were excluded from further analysis, we retained 8053 different metazoan organisms for subsequent analyses.

**Analysis and quantification of mtDNA rearrangements across metazoan evolution.** To study the evolutionary landscape of mtDNA gene organization changes during the evolution of Metazoa, a pairwise distance matrix was created using the common interval rearrangement explorer, CREx[53] (v1.0.0). The command-line interface version of CREx was run on each pairwise combination of organisms (8053 × 8053) with the following command "crex2 -f [path_to_file] -d -c", where "-d" returns only the distance and "-c" switches to the older and more established CREx1 algorithm. CREx defines distance as the most parsimonious (minimal) number of steps required to sort one organism's gene order from another while considering four possible rearrangement types: Translocations, reverse translocations, inversions, and tandem-duplication-random-loss events. The distance measurement chosen for our analysis is defined as the minimal number of breakpoints separating two gene orders. Briefly, breakpoints are defined as locations where a common interval ends, and the two gene orders differ[80]. To visualize the N x N distance matrix generated, a Python implementation of the

t-SNE algorithm was used, which is part of the scikit-learn[81] (v0.23.1) with a chosen perplexity value of 30, chosen because it yielded the most informative visualization.

**Codon usage and tRNA repertoire concordance measurements.** To identify organisms with a discrepancy between the codons recognized by their mitochondrial tRNA (mt-tRNA) repertoire and codon biases within their mtDNA protein-coding genes, all mtDNA protein-coding sequences from each tested organism were downloaded and isolated. Reading frames were validated by the presence of a starting codon on the 5′-end and a STOP codon on the 3′. The mitochondrial codons were translated based on taxon-specific mitochondrial translation tables (see NCBI Genetic Codes)[82]. In some organisms, the STOP codon was missing from certain mtDNA encoded sequences, as it is known to be added post-transcriptionally[83]. Therefore, in cases where the 3′-end of a gene coding sequence did not terminate with a STOP codon, a maximum of two A bases were artificially appended to the 3′-of the DNA sequence. If this did not result in the creation of a new STOP codon, the gene was considered erroneous and omitted from further analysis. Codon usage bias was measured using a Python implementation (CAI package, v1.0.3)[84] of relative synonymous codon usage (RSCU). Defined for each codon as:

$$RSCU = \frac{o_{ac}}{\left(\frac{1}{k_a}\right) \sum_{c \in C_a} o_{ac}} \quad (1)$$

Where $o_{ac}$ is the count of codon $c$ used by amino acid $a$, $c$ is the index for codons, $c_a$ is the set of codons used by a given amino acid and $k_a$ is the number of synonymous codons of amino acid $a$ (codon degeneracy)[85]. The codons covered by the mt-tRNA repertoire were determined based on each tRNA's anticodon sequence, which was reported in the relevant GenBank file and confirmed using tRNAScan-SE. The distributions of RSCU values for each amino acid, within each phylum, for codons recognized and unrecognized by the mt-tRNA were compared using a two-sided Mann–Whitney test and corrected for false discovery rate separately for each phylum, using the Benjamini–Hochberg method. We postulated that each mtDNA codon is recognized by a single tRNA. Notably, there are several examples of the recognition of degenerate codons in the same codon box by a single mt-tRNA[69,86]. Importantly, since not much is known about tRNA wobble recognition across evolution, our assumption allows for a coherent analysis across our database and can only underestimate actual codon recognition biases across evolution. Additionally, to reduce the effect of strand-specific and codon position-specific base composition biases on the results, we first calculated the base frequencies in protein-coding genes of all seven analyzed phyla separately for each strand and codon position combination. Then, we performed the above-described analysis again, while excluding the RSCU values of all codons that contain one or more bases that are the most prevalent for each strand and codon position per phylum. For each codon, we considered the genes it appears most commonly in, per strand.

**Calculation of mitochondrial genome architecture change rates and permutation tests.** To statistically assess the variability in mtDNA organization among metazoans, we calculated the mitochondrial genome architecture (AR) change rate for each taxonomic class, as previously performed[19]. AR rate is calculated as follows:

$$AR_{Rate} = \frac{N_{AR} - 1}{N_{mtDNA} - 1} \cdot 100 \quad (2)$$

Where $N_{AR}$ and $N_{mtDNA}$ are the number of different ARs and the total number organisms with sequenced mtDNA within that taxon, respectively. AR rates were calculated for all classes harboring at least 20 organisms. To generate distributions of expected and observed AR change rates, while considering large variations in sample sizes of the available metazoan classes (ranging from 21 in Ophiuroidea to 2471 organisms in Actinopterygii), we randomly sampled 21 organisms from each class 10,000 times using a custom R script and calculated AR change rate for each class with and without prior shuffling for the expected and observed populations respectively. The two distributions were visualized as the following ratio: $\frac{observed_{AR}}{expected_{AR}}$.

**Calculating the prevalence of all possible gene clusters and their frequencies.** To compare the evolutionary conservation of gene clusters, we performed an exhaustive search through all possible ordered lists of two or more genes for each available mtDNA gene order (MGO) in our database. We generated a non-redundant list of MGOs and measured the prevalence of each MGO within each available phylum. Since the mtDNA of most metazoan organisms is circular, the starting positions were arbitrarily reported. Therefore, to prevent inaccurate prevalence measurements due to variation in the predetermined genomic location of the mtDNA position 1, we anchored the gene orders of all mtDNAs with circular topology onto the gene ND1, which is present in almost all metazoans tested (99.5%). Additionally, we took circularity into account when required, by allowing for gene clusters that continue through position 1 of the gene order. For example, the gene pair A, C exists within a circular mitochondrial gene order C, B, D, E, A. To measure the proximity frequency of each pair considering only the 13-core protein-coding genes and 2 rRNAs, their pairwise prevalence as immediate

neighbors (on either strand) was measured for each phylum while ignoring tRNA genes. A gene pair was marked as a different strand pair if the two neighbors appeared on different strands in over 50% of the organisms within a given phylum. The heatmap visualization and hierarchical clustering were performed using the R package ComplexHeatmap (v2.9.3)[87]. The rows (gene pairs) and columns (phyla) were clustered using complete hierarchical clustering based on Euclidean distance.

**Processing of RNA-seq and PRO-seq data.** To generate a compendium of available RNA-seq samples from as many metazoans as possible, publicly available RNA-seq samples belonging to 98 different sequencing projects were downloaded from ENA as raw fastq files (Supplementary Data S2). The files underwent adapter trimming using Trim Galore[88] with the "—paired" parameter where needed and were subjected to quality control using a Phred cutoff score of 20. Firstly, the trimmed read files were non-uniquely mapped against the relevant organism mtDNA reference sequence using STAR aligner[89]. After mapping, a new, sample-specific, consensus reference sequence was generated using SAMtools[90] (v1.9.0), while using the mpileup command with the "-u" parameter to create a genotype likelihood file, followed by BCFtools[91] (v1.9.0) call command with the "-c" parameter to generate a consensus variant call format (VCF) file. The resultant VCF file was, in turn, converted into a standard FASTA-style reference sequence. Next, STAR was used to uniquely align the trimmed fastq files against the sample-specific reference. Following the second alignment, reads were quantitated irrespective of their strand, per position using SAMtools mpileup with a quality cutoff of 30, per gene using HTseq-count[92] (v0.11.2), and per intergenic junction using custom scripts and HTseq-count (further elaborated in the next section). Unlike the RNA-seq process, PRO-seq reads were only quantitated per position using the SAMtools mpileup command. Additionally, to account for the circularity of the mtDNA, a third alignment was performed only for PRO-seq samples, against the re-constructed sample-specific reference with the last 500 bases prepended to the reference. Mapping was performed and read coverage from the former circle junction was calculated as explained above and added to the previous mapping results. This approach allows reads that originate from the arbitrarily assigned mtDNA position 1 to map reliably to the correct position, resulting in increased sequencing reads coverage around the artificially assigned 5'- and 3'-end regions[20].

**Comparison of read density in same strand versus different strand gene–gene junctions.** We aimed towards calculating differences between RNA-seq reads encompassing gene-gene junctions encoded by the same strand to gene-gene junctions encoded by different strands. This approach was employed to identify organisms in which the end of precursor mitochondrial RNA transcripts in each strand (polycistron) maps to mtDNA positions encompassing strand switches between genes. Notably, this approach stems from the assumption that mtDNA transcription involves polycistrons, namely that precursor RNA transcripts encompass more than a single gene. To this end, custom annotation (gff3) files were generated for each organism's gene-gene junctions across the entire mtDNA sequence (apart from the junction between the first and the last genes, which often crosses a large non-coding locus). The analysis required that (A) the junction reads will be counted in a window designed to overlap either the 3' or 5' end of the gene and the intergenic region (IGR) with (B) a maximal length of 95% of twice the read size ($0.95 * 2l$) where $l$ is the average read length of a given sample. Then, HTseq-count was employed with the following parameters: "—mode intersection-strict", "—nonunique none", and "-s no". These parameters and the window size restrictions were designed to reduce the noise from reads originating from mature mRNA (which would only map to the gene body, and not to the intergenic region). In cases where the IGR between two genes was longer than twice the read size, the 'junction' was split into two windows, with one overlapping the edge of the 5' gene and the other overlapping the edge of the 3' gene; the overall junction coverage was then defined as the sum of both window counts. Both junctions and gene counts were then normalized using transcripts per million (TPM). TPM was calculated as follows:

$$TPM = \frac{reads}{reads_{total} \cdot length} \cdot 10^9 \tag{3}$$

Where *reads* and *length* are the read coverage and length of a given read/ junction and $reads_{total}$ is the sum of all reads in a given library.

**Identification of enriched motifs in insects with alternating gene block organization.** To detect motifs that are over-represented in arthropods that have alternating gene block organization, we used XSTREME[93] (v5.4.1), which combines motif discovery with motif enrichment analysis and clustering. XSTREME was run using the entire mtDNA sequences of all arthropods with alternating gene blocks organization as a primary input, against a negative control (background) of the mtDNA sequences of arthropods that do not have alternating gene block organization. The output was then analyzed using a Python script. We filtered the motif table for significantly enriched motifs that overlap with or are within less than 500 bases of DSJs in at least 80% of the analyzed species. The consensus motif was created using the STAMP web application, which generates a consensus sequence based on a pairwise alignment of all motifs[94]. To determine the protein level conservation score of *ND1*, we used Protein Residue Conservation Prediction[95] in Jensen–Shannon[96] scoring mode, with the amino acid sequences of *ND1* in all

sequenced insect species as input. The conservation score was then mapped against the DNA sequence of *ND1* in *D. melanogaster* for visualization.

**Detection of TTS and TIS sites in PRO-seq data.** To identify TTS and transcription initiation sites (TIS) across the mtDNA we used PRO-seq data from *Drosophila melanogaster* while employing a newly modified version of a previously designed approach[20]. TIS and TTS were defined as positions with a drastic increase or decrease, respectively, in the mean read coverage of a 500 bp downstream window in comparison to a 200 bp upstream window. The comparison was performed indirectly by comparing the read coverage in each window against the mean read coverage of the combined, 700 bp window multiplied by a scaling factor. Since the scaling factor determined the sensitivity of the algorithm, it was separately optimized for each sample, by starting at 0.9 and iteratively decreasing it by 0.05 until no read count peaks or valleys were called and selecting the minimum value that allows for peak detection. After the first iteration of peak and valley detection, local aggregations of peaks and valleys (within a 100 bp window) were combined into a single position, which was defined as the mtDNA position with maximum read coverage for peak aggregates, and the position with minimum read coverage for valley aggregates. Next, peaks and valleys were combined across multiple samples as follows: Firstly, we calculated a position-specific confidence score by:

$$Confidence\ Score = \frac{avg_{ds}}{avg_{ds} + avg_{us}} \tag{4}$$

for TIS and $1 - \frac{avg_{ds}}{avg_{ds} + avg_{us}}$ for TTS, where $avg_{ds}$ corresponds to the mean read coverage downstream in a 250 bp window, and $avg_{us}$ corresponds to the mean read coverage upstream in a 50 bp window. Lastly, neighboring positions identified as putative TIS and TTS were separately refined across different samples from the same experiment by changing their position to be the weighted average of all samples, with the weights defined as the confidence scores.

*Statistics and reproducibility.* Statistical analyses were performed using tests as indicated within each of the figure legends and along the main text. Standard deviations (SD) rather than standard errors were used, and confidence intervals were indicated where relevant. Sample sizes are indicated within the relevant figures (see Fig. 1 for example). Finally, a permutations test was used in the frame of the analysis presented in Fig. 2c, d and the number of iterations was indicated (shuffling 10,000 times), the entire process is described (see Methods).

**Reporting summary.** Further information on research design is available in the Nature Portfolio Reporting Summary linked to this article.

## Data availability
All sequence data analyzed in the study were downloaded from both the NCBI organelle database (https://www.ncbi.nlm.nih.gov/genome/organelle/) and the MitoZoa website (http://srv00.recas.ba.infn.it/mitozoa/). RNA-seq experimental data were downloaded according to accession numbers listed in Supplementary Data S2. All data used to generate the figures throughout this work and the supplementary data are available in FigShare. https://figshare.com/projects/Shtolz_2022_mtDNA_evolutionary_rearrangements/156008

## Code availability
All code used for the analyses and visualizations in the manuscript is available on GitHub: https://github.com/Noam-St/2022_Metazoan_mtDNA_Project

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

## Acknowledgements

This work was funded by the Israeli Science foundation grants 372/17 and 404/21 and by the US Army Research Office grants LS67993 and LS80581 awarded to D.M. The authors are also funded by Myles Thaler Chair in Genetics and Genomics Research (D.M.).

## Author contributions

N.S. performed all the analyses, generated all the figures, and participated in writing the manuscript; D.M. conceived the study and wrote the manuscript.

## Competing interests

The authors declare no competing interests.
