## [Peer Review File · Communications Biology]

Reviewers' comments:

Reviewer #2 (Remarks to the Author):

This paper describes a possible relationship between mitochondrial gene organization and transcription units in metazoans. The authors suggested that in arthropod mtDNA, such as *Drosophila*, the junction between the gene encoded on one of the two strands and the other strand tends to be less transcribed than the other region. This suggests to the authors that there are transcription start and end points in these regions, although this has not been suggested in the vertebrate mitochondrial genome. The authors' analysis is extensive and should provide a useful perspective for readers interested in the structure of metazoan mtDNA and its evolution. The organization of the metazoan mtDNA gene arrangement, gene composition, etc. in this paper will also provide useful information to readers interested in the structure and evolution of metazoan mtDNA. I was intrigued by these authors' analysis.

However, there are several questions/comments on authors' analyses as follows.

General questions.

1. However, if authors' view on the relationship between transcriptional brocks and gene organizations, are there any sequence (or structural) motives for transcriptional initiation and termination? In addition, the junctions of different strand encoded genes seem not to match the transcriptional start points and termination points. In echinoderm mitochondrial genomes, multiple transcription starting points have been suggested. Does it match authors' results? I would like to know this issue.

2. The process of transcription of metazoan mitochondrial genomes seems to overlap with the process of replication of metazoan mitochondrial genomes. Long noncoding regions typically found in vertebrate and arthropod mitochondrial genomes are known to contain initiation regions of both replication and transcription. I think that authors are needed to analyze/discuss their data with relation of replication process. For example, in Fig. 5B, transcription of "L strand" in the long noncoding region is shown, but there are no mentions in this paper. Is transcription on this region biologically less important?

In addition to that, the sense strand of all (or almost all) metazoan mitochondrial genomes in which all genes are encoded on one strand is the H strand (at least T>A). This includes mitochondrial genomes with a high frequency of gene rearrangements, such as most nematode mitochondrial genomes and urochordate (tunicate) mitochondrial genomes. These appear to suggest a link between mitochondrial genome structural variability and replication.

Further questions/comments are as follows.

1. The paragraph starting from 5th line from bottom of P. 5.

The authors define an mtDNA codon as a codon that has a base in the third letter of the codon that forms a Watson-Crick base pair with the base in the first letter of the mt-tRNA anticodon. Based on this definition, the frequency of amino acids specified by mtDNA codons is high in vertebrate mitochondrial genomes, moderate in arthropod/mollusk mitochondrial genomes, and low in nematode mitochondrial genomes.

However, I believe this summary is a misunderstanding. The main coding strand of vertebrate mitochondrial genomes is the L strand, where A and C occur more frequently in the third letter of the codon; the ND6 gene encoded on the H strand is shorter (this gene often has T and G in the third letter of the codon). This bias in codon usage frequency can be attributed to a strand-specific base composition bias due to non-contrastive replication methods of the mitochondrial genome, for example. On the other hand, the anticodon of metazoan mt-tRNA is U or G in the first letter of the anticodon, with very few exceptions, except for the first letter of the anticodon of tRNA-Met, which is C. This is true regardless of the different phyla of metazoans, including vertebrates, arthropods, and linear animals. The main coding strand of nematode mtDNA is the H strand, which is inverted from

vertebrate mtDNA. Arthropod mtDNAs have averaged codon usage frequencies, as both strands encode protein genes of similar length. Among chordates, the cephalochordate mtDNA is almost identical to the vertebrate mtDNA in gene arrangement, but the main coding strand is the T- and G-rich H strand, and the "mtDNA codon" as defined by the authors is used less frequently. Based on the above, I find the authors' discussion of tRNA and codon usage frequency and related issues unacceptable, and I urge them to completely reanalyze and rethink the issue.

2. Fig. 5B

Authors indicated that the major coding strand of *Drosophila* mtDNA is H strand and the other is L strand. However, definition of H strand and L strand is difference of density of two strands caused by the difference of nucleotide composition between two strands. H strand is rich in T and G rather than A and C, and L strand is rich in A and C. As the case of vertebrate mitochondrial genome, the strand coding COI gene (noted as H strand in this figure) is L strand. H strand is the template strand of COI gene, for instance. I think that authors' definition of L and H strands are unlikely. Authors should correct them though this paper.

Minor comment.

1. Fig. 1E: The Fig. 1E seems to be broken. Authors should correct this figure.

Reviewer #3 (Remarks to the Author):

Shtolz & Mishmar present an extremely useful paper which summarizes and tests a lot of 'accepted wisdom' around mt genome dynamics. It is valuable for these phenomena to be put together in a solid meta-analysis that points the way forward for our field even if these are results that have been discussed for over 20 years at this point (e.g. phylum level MGO conservation, 'special' gene pairs like *atp6-8*, bias towards tRNA codon use etc.). The additional work on transcriptional analysis for the basis of the constraints demonstrated makes it any even more valuable work.

It is well analysed, the methods are solid for others to work up further and the figures great for explaining the authors points. I have only a few minor questions for clarification, but expect to see this published more or less as written shortly.

1) Could the author comment a little further on tRNAs which are inverted relative to the gene block in which they fall e.g. Q or C-Y in *Drosophila* (and the vast majority of arthropods)? What do these genes look like within your transcription analysis methods, and what does that say for the seeming inefficiency of having to transcribe the opposite strand version of a block for 1 or a handful of tRNAs.

2) I was a bit confused by the distinction between taxa with alternating and non-alternating gene blocks. Even vertebrates have genes transcribed on both strands and they are arranged in blocks (ND6 vs everything else) - so the distinction seems one of degree (more blocks in arthropods vs less in vertebrates). Or is there a clear transcriptional distinction between these categories that can be demonstrated by the run-on methods, and then inferred for non-model species by the transcriptional read analysis performed here? Sorry if I missed something but it might improve the paper to tighten up this distinction.

Otherwise fine work.

Below please find our point-by-point replies to the comments raised by the two referees:

Reviewer #2

1. If authors' view on the relationship between transcriptional blocks and gene organizations, are there any sequence (or structural) motifs for transcriptional initiation and termination?

> We agree that the relationship between a specific alternating gene block organization and transcription implies that some regulatory elements, such as transcription factors, are likely to recognize and bind in the regions along the edges of the blocks, and in turn modulate mtDNA transcription initiation and/or termination. Therefore, as a first step to address this comment we searched for sequence motifs that are enriched only in arthropods with alternating gene block organization and are within or close to junctions surrounded by genes that are in different strands (DSJ). Our analysis revealed five such motifs which are conserved among arthropods; this initial step enabled the identification of a consensus sequence motif that is shared by the DSJ and is A/T rich like the mtDNA binding site of the drosophila dmTTF (mitochondrial transcription termination factor). These findings are summarized in the Results section, page 10, in a new subchapter entitled '**Different strand junction loci are enriched for motifs in species with alternating gene block organization**':

"If gene blocks mark the boundaries of mtDNA transcriptional units in arthropods, it is expected that such boundaries will be bound by transcription factors that will mediate transcriptional initiation from the one end, and termination from the other end. In humans, it has been discovered that the transcription factors TFB2M, POLRMT, and TFAM form the core of the mitochondrial transcription initiation complex, while mTERF1 (and possibly additional members of the mTERF family) mediate transcriptional termination (58). Although protein orthologues for these factors have been identified in multiple metazoans (59-61), little is known about the function of the mitochondrial transcriptional machinery in non-human species, especially invertebrates, including arthropods. As a first step towards a mechanistic analysis of the borders of arthropod transcriptional units, we hypothesized that species with alternating mtDNA gene block organization provide an excellent opportunity to identify sequence motifs within the block-block boundaries. Therefore, we searched our metazoan database for sequence motifs enriched in the regions of block-block boundaries in arthropods with alternating gene block organization. Specifically, we screened for sequence motifs located no more than 500 bases away from either side of the DSJ. We found a total of 39 such motifs that were significantly enriched in species with alternating gene block organization (FDR-corrected E-value < 0.05). Five of these motifs were near DSJs (Fig 6A), three within tRNA genes between ND3 and ND5, one between trnP and trnT, and another motif within ND1, yet 378 bases away from the DSJ between ND1 and CYTB (Fig 6B). Importantly, the ND1 motif was identified in a relatively non-conserved region at the protein level, yet conserved at the DNA level, which supports its possible regulatory

function (Fig 6C). Notably, dmTTF, an mTERF-like termination factor has been shown to bind intergenic junctions in *Drosophila* mtDNA in-vitro and to recognize AT-rich motifs (61). Most interestingly, we found three sequence motifs (indexed as motifs 2, 4 and 5) that are all within 50 bases of the putative binding region and 300 bases of the predicted transcription termination position. Additionally, these five motifs form an AT-rich consensus sequence which is 54.5% similar to the reported dmTTF binding sequence in *Drosophila*. Taken together, these results provide an initial basis for the functional mechanism underlying the alternating gene block organization. Taken together, these results provide an initial basis for the functional mechanism underlying the alternating gene block organization”.

2. The junctions of different strand encoded genes seem not to match the transcriptional start points and termination points. In echinoderm mitochondrial genomes, multiple transcription starting points have been suggested. Does it match the authors’ results?

> We thank the reviewer for this comment. It is important to clarify, that our attempt to detect the polycistronic units of as many species as possible using RNA-seq, does not provide a sufficiently high resolution to locate the exact sites of transcriptional initiation and termination in a base-pair accuracy. Accordingly, we do not claim that all DSJs in each species should exactly overlap with the polycistronic borders. In fact, we clearly show that in the vast majority of chordates, DSJ with *ND6* - the only protein-coding gene in the reverse strand, does not mark the border of polycistrons. However, we do claim that DSJs are in correlation with the borders of mtDNA transcriptional units in species with alternate gene blocks organization, i.e., most arthropods. This is further supported by PRO-seq and RNA-seq analysis of *Drosophila*. As most arthropods have a similar mtDNA gene organization to *Drosophila*, and lower expression in DSJs (again – as in *Drosophila*) the correlation between gene block organization and transcriptional units’ organization is likely conserved in arthropods. This is very different from Echinoderms: the mtDNA organization of Echinoderms does not follow alternating gene block organization and hence does not have an arthropod-like DSJ expression pattern. We do agree that higher resolution analysis of mtDNA transcription across evolution, including Echinodermata, should be performed in the future. This point has been added to the Discussion section (page 12):

“Taken together, one cannot easily deduce from the observed mtDNA transcriptional pattern in a certain organism to the entire metazoan phylogeny. Thus, future analyses of mtDNA transcription across the metazoan phylogeny are imperative.”

3. The process of transcription of metazoan mitochondrial genomes seems to overlap with the process of replication of metazoan mitochondrial genomes. Long noncoding regions typically found in vertebrate and arthropod mitochondrial genomes are known to contain initiation regions of both replication and transcription. I think that authors are needed to analyze/discuss their data concerning the replication process. For example, in Fig. 5B, transcription of “L strand” in the long noncoding region is shown, but there are no mentions in this paper. Is transcription in this region biologically less important? In addition to that, the sense strand of all (or almost all) metazoan mitochondrial genomes in which genes are all encoded on one strand is the H strand (at least T>A). This includes

mitochondrial genomes with a high frequency of gene rearrangements, such as most nematode mitochondrial genomes and urochordate (tunicate) mitochondrial genomes. These appear to suggest a link between mitochondrial genome structural variability and replication.

>We agree that there are most likely a variety of functional consequences for mtDNA rearrangements, which are beyond transcription yet beyond the scope of this study. Nevertheless, while it is true that in mammals, transcription, and replication initiate from proximal regions in the non-coding region and that these mechanisms are likely dependent, to our knowledge this tight connection was not experimentally studied in other metazoans. Furthermore, the multiple transcriptional initiations and termination sites across the entire *Drosophila* mtDNA (including coding sequences), as we show in the current manuscript and previously (REF20 - Blumberg et al 2017, *Genome Research*), challenge the connection between replication and transcription in *Drosophila*, and probably in all other organisms with alternating gene block organization. To better communicate this point in the paper, we referred to it in the Discussion (page 12):

“Nevertheless, one cannot rule out the functional involvement of factors other than transcription. Indeed, mtDNA transcription is known to be heavily coupled to replication in mammals, with both processes relying on the same RNA polymerase, used for both primer synthesis and transcription (58, 72). However, organisms with alternating gene block organization like *Drosophila* contain additional initiation points, outside of the mitochondrial non-coding region and the origin of replication, and thus contain transcription initiation points that are most likely decoupled from replication. Taken together, one cannot easily deduce from the observed mtDNA transcriptional pattern in a certain organism the entire metazoan phylogeny. Thus, future analyses of mtDNA transcription across the metazoan phylogeny are imperative.”

4. The paragraph starting from the 5th line from bottom of P. 5.
The authors define an mtDNA codon as a codon that has a base in the third letter of the codon that forms a Watson-Crick base pair with the base of the first letter of the mt-tRNA anticodon. Based on these definitions, the frequency of amino acids specified by mtDNA codons is high in vertebrate mitochondrial genomes, moderate in arthropod/mollusk mitochondrial genomes, and low in nematode mitochondrial genomes.
However, I believe this summary is a misunderstanding. The main coding strand of vertebrate mitochondrial genomes is the L-strand, where A and C occur more frequently in the third letter of the codon; the ND6 gene encoded on the H strand is shorter (this gene often has T or G in the third letter of the codon). This bias in codon usage frequency can be attributed to strand-specific base composition bias due to non-contrastive replication methods of the mitochondrial genome, for example. On the other hand, the anticodon of metazoan mt-tRNA is U or G in the first letter of the anticodon, with very few exceptions, except for the first letter of the anticodon of tRNA-Met, which is C. This is true regardless of the different phyla of metazoans, including

vertebrates, arthropods, and linear animals. The main coding strand of nematode mtDNA is the H strand, which is inverted from vertebrate mtDNA. Arthropod mtDNAs have averaged codon usage frequencies, as both strands encode protein genes of similar length. Among chordates, the cephalochordate mtDNA is almost identical to the vertebrate mtDNA in gene arrangement, but the main coding strand is the T- and G-rich H strand, and the "mtDNA codon" as defined by the authors is used less frequently. Based on the above, I find the authors' discussion of tRNA and codon usage frequency and related issues unacceptable, and I urge them to completely reanalyze and rethink the issue.

>We certainly agree that mtDNA base composition in each strand is a factor that we should consider in our analysis. Firstly, we also agree that the strand nomenclature is confusing since many papers clearly state that in humans the H strand is identical to the main coding sequence translated from RNA (including the original Cambridge Reference sequence)^{1,2}, while other papers argue otherwise (see citation 51)³. However, this is certainly not the main issue of our manuscript. Therefore, to avoid confusion we choose to refrain from using the terms 'heavy' strand and 'light' strand which were coined based on the human and mice nomenclature and used the terms 'forward' and 'reverse' strands instead. This is now conveyed in the Results section (page 6):

"In mammals, protein-coding mtDNA genes are known to exhibit base composition biases, which are exhibited by both strand asymmetry (47) and position within the codon (48). We thus reasoned that such biases may confound our results. Prior to correcting for such biases, it is important to note that the heavy and light strand terminology of the mtDNA is a source of confusion and is largely based on the mtDNA nomenclature of mice (49) and human (50): the heavy strand has a high G + T content as compared to the light strand and is the so-called 'coding' strand (51). However, that is not the case for other phyla, such as nematodes, in which all genes are located in a single strand and arthropods which do not have any reported coding strand asymmetry (53). Therefore, to avoid inaccuracies, we choose to refrain from this terminology throughout the paper and name the strands based only on their directionality (forward and reverse). This nomenclature will be used for our correction for base composition biases (see below)."

Secondly, we calculated base composition per mtDNA strand in all tested taxa and found that there is variability in the nature of this trait. Therefore, in the analysis that we performed, we accounted for base composition biases per strand, codon position, and phylum. Our findings revealed that the trend towards usage of mtDNA-encoded tRNAs was retained, as described in the Results section (page 6) and accompanying figures (Supplementary figures S2 + S3):

"In order to consider possible base composition biases between the strands, and per codon position, we first assessed such biases across metazoans (Supplementary Fig S2). Then, we re-performed our analyses while taking this parameter into account, while

focusing only on codons that do not contain the most prevalent base within each codon position, per strand, and per phylum. Notably, while this stringent filtration unsurprisingly removed most codons, 9 amino acids out of 19 (47%) retain a significant mt-tRNA codon preference in Chordata and 6 out of 10 (60%) in Arthropoda. Additionally, none of the amino acids that have significant biases changed the bias directionality due to our filtration (Supplementary Fig S3, as compared to S1). Overall, these results suggest a preference for the usage of mt-tRNAs for subsequent translation of mtDNA-encoded PCGs in a phylum-specific manner.”

This calculation is further elaborated on in Methods (page 15):

“To reduce the effect of strand-specific and codon position-specific base composition biases on the results, we first calculated the base frequencies in protein-coding genes of all seven analyzed phyla separately for each strand and codon position combination. Then, we performed the above-described analysis again, while excluding the RSCU values of all codons that contain one or more bases that are the most prevalent for each strand and codon position per phylum. For each codon, we considered the genes it appears most commonly in, per strand.”

5. Fig. 4B

Authors indicated that the major coding strand of *Drosophila* mtDNA is H strand and the other is L strand. However, definition of H strand and L strand is difference of density of two strands caused by the difference of nucleotide composition between two strands. H strand is rich in T and G rather than A and C, and L strand is rich in A and C. As the case of vertebrate mitochondrial genome, the strand coding COI gene (noted as H strand in this figure) is L strand. H strand is the template strand of COI gene, for instance. I think that authors’ definition of L and H strands are unlikely. Authors should correct them though this paper.

>We agree that the H and L strand terminology, which was previously coined based on the analysis of human and mouse mtDNAs, may lead to confusion while performing a large evolutionary analysis. As stated in our reply to the previous comment, to avoid confusing strand nomenclature we used the terms ‘forward’ and ‘reverse’ for the two mtDNA strands, as conveyed in the Results section (page 6):

“Prior to correcting for such biases, it is important to note that the heavy and light strand terminology of the mtDNA is a source of confusion and is largely based on the mtDNA nomenclature of mice (49) and human (50): the heavy strand has a high G + T content as compared to the light strand and is the so-called ‘coding’ strand (51). However, that is not the case for other phyla, such as nematodes, in which all genes are located in a single strand and arthropods which do not have any reported coding strand asymmetry (53). Therefore, to avoid inaccuracies, we choose to refrain from this terminology throughout the paper and name the strands based only on their directionality (forward and reverse).

6. Fig. 1E: The Fig 1E seems to be broken. Authors should correct this figure.

>Corrected.

Reviewer #3

1. Could the authors comment a little further on tRNAs which are inverted relative to the gene block in which they fall e.g., Q or C-Y in *Drosophila* (and the vast majority of arthropods)? What do these genes look like within your transcription analysis methods, and what does that say for the seeming inefficiency of having to transcribe the opposite strand version of a block for 1 or handful of tRNAs.

>Unfortunately, our RNA-seq-based transcription unit analysis methods do not reliably capture tRNA-only transcriptional units, since short and non-coding RNA genes such as tRNAs are lost in most RNA-seq protocols, which use longer reads. However, we can partially address this comment based on our analysis of available *D. melanogaster* PRO-seq data, which is not limited by the constraints mentioned for RNA-seq. Accordingly, at least in the case of *Drosophila*, these genes are co-transcribed in a single polycistron, without “wastefully” transcribing long portions of non-coding RNA. This seemingly inefficient mechanism is undoubtedly more economical than the vertebrate transcriptional scheme, which encompasses nearly the entirety of the mtDNA. Since these tRNA genes remain proximal across many different arthropods, we can speculate they are co-transcribed in other arthropod species. However, it would certainly be of great interest to perform PRO-seq on additional Arthropoda species in the future. This point was added to the Discussion section (page 12):

“Taken together, one cannot easily deduce from the observed mtDNA transcriptional pattern in a certain organism to the entire metazoan phylogeny. Thus, future analyses of mtDNA transcription across the metazoan phylogeny are imperative.”

2. I was a bit confused by the distinction between taxa with alternating and non-alternating gene blocks. Even vertebrates have genes transcribed on both strands and they are arranged in blocks (ND6 vs everything else) - so the distinctions seem one of degree (more blocks in arthropods vs less in vertebrates). Or is there a clear transcriptional distinction between these categories that can be demonstrated by the run-on methods, and then inferred for non-model species by the transcriptional read analysis performed here? Sorry if I missed something but it might improve the paper to tighten up this distinction.

>We thank the referee for pointing out a possible lack of clarity in our definition of the alternating gene block organization of the mitochondrial genome. The latter is illustrated in Figure 7 of the manuscript. Note that the so-called alternate gene block organization is common in arthropods, whereas the organization with a single protein-coding gene (ND6) in the opposite strand is common in vertebrates. Furthermore, the

distinction between the types of mtDNA organization is first and foremost based on the organization of non-tRNA genes and initially defined based on the organization observed in *Drosophila melanogaster*. Species with an alternating gene block organization contain clusters of 2 or more co-oriented non-tRNA genes that continuously alternate between the two mtDNA strands. This organization is distinct from the typical vertebrate organization based on our definition of “blocks” as groups of 2 or more genes. To better communicate this point, we rephrased the text on page 10 of the Results section:

“While inspecting mtDNA gene orders, we noticed that ~90% of the arthropod species display an mtDNA organization that is like in *Drosophila*: i.e., co-oriented groups of two or more non-tRNA genes that continuously alternate between the two mtDNA strands. We henceforth term this organization alternating gene block organization”.

References

- 1 Andrews, R. M. *et al.* Reanalysis and revision of the Cambridge reference sequence for human mitochondrial DNA. *Nature genetics* **23**, 147 (1999).
<https://doi.org:10.1038/13779>
- 2 Rackham, O. & Filipovska, A. Organization and expression of the mammalian mitochondrial genome. *Nature Reviews Genetics*, 1-18 (2022).
- 3 Alexeyev, M. Mitochondrial DNA: the common confusions. *Mitochondrial DNA Part A* **31**, 45-47 (2020).

REVIEWERS' COMMENTS:

Reviewer #2 (Remarks to the Author):

I think the revised version of this manuscript is improved by the extensive responses to the reviewers' comments. I think it is well organized for the coevolution of mitochondrial genome structure and function and will be useful to readers who are considering this issue.

I would like to address a few points that remained in my mind. In "Codon usage and tRNA repertoire concordance measurements" in "Methods", the authors state "Notably, for the sake of simplicity, we postulated that each mtDNA codon is recognized by a single tRNA, and that each tRNA can recognize a single codon.". The first half of the assumption is correct, but the second half, already stated, is difficult to understand. It would be more accurate to say that each codon is recognized by a single tRNA, and that degenerate codons in the same codon box are recognized by the same tRNA.

Reviewer #3 (Remarks to the Author):

The revised manuscript thoroughly addresses the prior reviewer comments in my opinion. I have nothing further to advise.

Below please find our point-by-point replies to the comments raised by referee #2:

Reviewer 2 wrote:

“I would like to address a few points that remained in my mind. In "Codon usage and tRNA repertoire concordance measurements" in "Methods", the authors state "Notably, for the sake of simplicity, we postulated that each mtDNA codon is recognized by a single tRNA, and that each tRNA can recognize a single codon.". The first half of the assumption is correct, but the second half, already stated, is difficult to understand. It would be more accurate to say that each codon is recognized by a single tRNA, and that degenerate codons in the same codon box are recognized by the same tRNA.”

>To address the comment raised the text was revised as follows: “We postulated that each mtDNA codon is recognized by a single tRNA. Notably, there are several examples for the recognition of degenerate codons in the same codon box by a single mt-tRNA (69,86). Importantly, since not much is known about tRNA wobble recognition across evolution, our assumption allows for a coherent analysis across our database and can only underestimate actual codon recognition biases across evolution.”